# Voices from the front lines: A qualitative study of integration of HIV, tuberculosis, and primary healthcare services in Johannesburg, South Africa

Naomi Lince-Deroche[1]*, Rahma Leuner[1], Sharon Kgowedi[1], Aneesa Moolla[1‡], Sinethemba Madlala[1‡], Pertunia Manganye[1‡], Barbara Xhosa[1‡], Caroline Govathson[1], Takiyah White Ndwanya[1], Lawrence Long[2]

1 Faculty of Health Sciences, Health Economics and Epidemiology Research Office, Department of Internal Medicine, School of Clinical Medicine, University of the Witwatersrand, Johannesburg, South Africa,
2 Department of Global Health, School of Public Health, Boston University, Boston, Massachusetts, United States of America

☉ These authors contributed equally to this work.
‡ These authors also contributed equally to this work.
* naomi.lince.deroche@gmail.com

**Data Availability Statement:** Data cannot be shared publicly because the local ethics committee restricts access. Data are available from the Health

## Abstract

### Introduction

In South Africa, in 2013–2014, provision of antiretroviral treatment (ART) shifted in some areas from NGOs to public facilities. Tuberculosis (TB) management has also been integrated into public services. We aimed to explore the opinions and experiences of service managers and healthcare providers regarding integration of HIV and TB services into primary healthcare services.

### Methods

The study sites included three clinics in one peri-urban/urban administrative region of Johannesburg. From March 2015 to August 2016, trained interviewers conducted semi-structured interviews with purposively selected participants. Participants were eligible if they were city/regional managers, clinic managers, or healthcare providers responsible for HIV, TB, non-communicable diseases, or sexual and reproductive health at the three study sites. We used a grounded theory approach for iterative, qualitative analysis, and produced descriptive statistics for quantitative data.

### Results

We interviewed 19 individuals (nine city/regional managers, three clinic managers, and seven nurses). Theoretical definitions of integration varied, as did actual practice. Integration of HIV treatment had been anticipated, but only occurred when required due to shifts in funding for ART. The change was rapid, and some clinics felt unprepared. That said, nearly all respondents were in favor of integrated care. Perceived benefits included comprehensive

Economics and Epidemiology Research Office (HE2RO (information@heroza.org) after obtaining approval the Health Research Ethics Committee (medical) at the University of the Witwatersrand. The study was protocol M140104. Permission to use the data can be obtained through a request to the Committee. Contact details are as follows: Research Office Secretariat, Faculty of Health Sciences, Phillip Tobias Building, 3rd Floor, Office 301, 29 Princess of Wales Terrace, Parktown 2193 Johannesburg, South Africa (Telephone: +27 (0)11 717 1252, Email: hrec-medical. researchoffice@wits.ac.za, Web: https://www.wits. ac.za/research/).

**Funding:** This study was made possible by the generous support of the American people through the United States Agency for International Development (USAID), award number AID-674-A-12-00029 to LL. The contents are the responsibility of the Health Economics and Epidemiology Research Office, a Division of the Wits Health Consortium (Pty) Ltd and do not necessarily reflect the views of USAID or the United States Government. LL received the grant.

**Competing interests:** The authors have declared that no competing interests exist.

case management, better client-nurse interactions, and reduced stigma. Barriers to integration included staff shortages, insufficient training and experience, and outdated clinic infrastructure. There were also concerns about the impact of integration on staff workloads and waiting times. Finally, there were concerns about TB integration due to infection control issues.

## Discussion

Integration is multi-faceted and often contingent on local, if not site-specific, factors. In the future in South Africa and in other settings contending with health service reorganization, staff consultations prior to and throughout phase-in of services changes could contribute to improved understanding of operational requirements, including staff needs, and improved patient outcomes.

## Introduction

Early in the HIV epidemic, HIV services were provided in vertical programs as a result of disease-specific funding or approaches to service provision [1]. However, segregated service provision was thought to result in patients needing to visit different facilities for their different health problems or needs, or having to visit the same facility on different days of the week [2]. This resulted in "missed opportunities" for addressing patients' holistic healthcare needs [1,3]. From a health system perspective, it also led to fragmented and inefficient service delivery and possibly duplication of services [4].

In response, in the early 2000s policy makers globally committed to "integration" of health services, focusing initially on integration of HIV and contraceptive services [5,6]. Over time, the discussion has shifted to integration of HIV and tuberculosis (TB) services as well as integration of HIV and TB care into primary healthcare services [7–9]. Globally, the definition of integration may vary depending on the services and location. It may refer to services provided at the same facility by different providers or by the same provider during the same client-provider interaction.

In South Africa, integration has been a key policy focus for many years. Integration as a concept is included in several national-level policy documents, and is seen as an important approach for offering efficient and comprehensive care in the country's public health system [10–15]. The National Department of Health initiated "re-engineering" of its public primary healthcare services in 2010, focusing on improving health outcomes and improving health system efficiency [16]. Shortly thereafter, it launched the country's plan for integrated chronic disease management, emphasizing the importance of, "integration of care for patients with chronic communicable and non-communicable diseases such that patients are treated as individuals and not disease entities" [17].

Integration has been particularly important for HIV care and treatment, TB management and sexual and reproductive health services in the country. South Africa has the greatest number of HIV-infected individuals and the second highest TB incidence rate worldwide [18]. Approximately 57% of all active TB patients are HIV co-infected [19]. When antiretroviral therapy (ART) and wide scale counselling and testing were initially made available in the country, local NGOs were supported by the United States' President's Emergency Plan for AIDS Relief (PEPFAR) to provide ART in standalone facilities referred to as Comprehensive

Care, Management and Treatment of HIV and AIDS sites, or CCMTs [20]. However, the South African government quickly took on more responsibility for care and treatment, and standalone facilities were folded into public health facilities throughout 2013–2014. In 2015, 78% of HIV-related care and treatment activities were funded by the South African government [21].

Today, nurse-initiated and managed HIV care and treatment is the norm in public facilities nationwide [22]. Integrated management of HIV, TB, and other primary care services, including sexual and reproductive healthcare services, is the vision in these settings. However, there is limited documentation of where and how integration is actually taking place. There has also been very little focus on the role of service managers and healthcare providers during and after the transition to integrated care. In this study, we aimed to explore the opinions and experiences of service managers and healthcare providers regarding integration of HIV and TB services into primary healthcare services, including why integration happened, what it looks like now, and possible limitations to its implementation and impact.

## Materials and methods

### Study population

The City of Johannesburg is divided into seven administrative regions. Each region has responsibility for providing a range of public services including healthcare. We used purposive sampling to select one region and, within the region, three of ten primary health clinics. Selection was done in partnership with the services managers for the administrative regions and was based on proximity to the researchers as well as the facilities' recent experiences with integration and openness to collaborating for research purposes. Two of the study clinics were located in a densely-populated, under-serviced peri-urban area. The third clinic, which was located about twenty minutes away from the other two clinics, was in an urban area.

We purposively selected the study participants from lists of eligible individuals at the city, region, and clinic levels. Managers employed by the City of Johannesburg working at the city's headquarters or the regional headquarters were eligible for participation in the study if they were responsible for service delivery, organization, or record keeping at the study clinics and directly or indirectly involved in the oversight of HIV, TB, non-communicable diseases (NCDs), or sexual and reproductive health (SRH) services. We also selected the three clinic managers. Finally, we selected healthcare providers from lists at each facility denoting staff who were routinely present at the clinics and who were directly or indirectly responsible for HIV, TB, NCDs, or SRH services.

### Data collection

We conducted interviews with the selected participants between March 2015 and August 2016. We used semi-structured interview guides, with quantitative and qualitative sections, developed for each target group (i.e. city/regional managers, clinic managers, and healthcare providers). (See S1 File). All three interview guides contained a core group of questions about the presence of integration and participants' opinions on the value of integration as well as unique sections with questions appropriate for each specific target group. Unique issues discussed with city and regional managers included historical and ongoing partnerships with NGOs for provision of services within the city, communication regarding policies, implementation of policies, and budgeting and resource planning for health services. The interviews with clinic managers and healthcare providers included questions on the services offered at the facilities, how services were staffed, the number of patients served, patient tracking, and training received by clinical staff. Clinic managers, like regional and city managers, were also asked

about communication regarding policies, implementation of policies, and budgeting and resource planning for health services.

Written informed consent was obtained from all participants. Ethical approval was obtained from the Human Research Ethics Committee at the University of Witwatersrand (protocol number M140104). Approval to conduct the study was also obtained from the study clinics and both the regional and city authorities in the City of Johannesburg.

## Analysis

We digitally recorded and transcribed each interview. Qualitative data were analyzed following a grounded theory approach in NVivo qualitative data analysis software (QSR International Pty Ltd., Version 11). Grounded theory assumes that the theories or explanations derived from a particular dataset are "grounded" in the data rather than preconceived prior to analysis [23,24]. We established a draft codebook based on preliminary analysis. Common themes within the data were then identified, and the codebook was updated iteratively to allow for an inductive approach to the analysis. Each interview transcript was coded by at least two members of the research team. Coding discrepancies were discussed until consensus was reached. Inter-coder reliability was formally assessed using Nvivo's built-in functionality, which led to further refinements in coding approaches and strengthening of inter-coder reliability.

Quantitative data were entered into REDCap data management software [25]. We then analyzed the data descriptively using Stata (Release 14. College Station, TX: StataCorp LP). Categorical variables are presented as frequencies and proportions. Continuous variables are presented as means and standard deviations. All proportions and means represent non-missing responses.

## Results

### Participant characteristics

We interviewed a total of 19 individuals: nine managers at the city or regional level, three clinic managers, and seven healthcare providers (all nurses) within the facilities. There were no refusals during the recruitment process. As noted in Table 1, the clinic-based participants represented 100% of all clinic-level managers and 21.2% of the day-to-day clinical healthcare providers (all of whom were nurses) at the three facilities. At the regional and city level, the nine managers represented all senior staff responsible for delivery of HIV, TB, and SRH services. Across all interviewee groups, years of experience working with the Department of Health increased as the level of seniority or responsibility for service delivery or management increased.

South Africa's "Core Norms and Standards" dictate the kinds of services that should be offered at the primary care level [26]. Clinic managers considered themselves to be indirectly responsible for all services dictated by the Core Norms and Standards. Considering the healthcare providers, all reported being responsible for offering chronic disease management services, and six of the seven healthcare providers reported responsibility for HIV and SRH services.

Table 2 describes staffing and service provision as reported for the three clinics. All three clinics routinely relied on nurses for a range of service provision. Doctors also provided services, though the doctors were not based at the facilities. A range of primary healthcare services were offered at the sites, including HIV treatment. Nearly all services were offered every day.

**Defining integration.** The study respondents were asked to comment on their understanding of integration, and how it should be defined and implemented in public health facilities. The city/regional managers outlined that when services are integrated, patients should not

**Table 1. Characteristics of study participants and responsibilities for health services provision in an administrative region of Johannesburg, South Africa (n = 19).**

| | City/regional managers (n = 9) | Clinic managers (n = 3) | Healthcare providers (n = 7) |
|---|---|---|---|
| Proportion of total individuals in category who participated in study, % (n/N) | - -[a] | 100% (3/3) | 21.2% (7/33)[b] |
| Years of experience at Department of Health, (mean [SD]) | 18.5 [8.8] | 4.2 [2.5] | 1.4 [0.3] |
| Responsible for provision of . . ., (n (%))[c] | | | |
| Chronic disease management | -- | 3 (100.0) | 7 (100.0) |
| HIV/AIDS | -- | 3 (100.0) | 6 (85.7) |
| Women's health/reproductive health | -- | 3 (100.0) | 6 (85.7) |
| Tuberculosis | -- | 3 (100.0) | 4 (57.1) |
| Immunizations/child health | -- | 3 (100.0) | 4 (57.1) |
| Mental health | -- | 3 (100.0) | 0 (0.0) |
| Medical male circumcision | -- | 0 (0.0) | 0 (0.0) |

SD = standard deviation.

Chronic disease management = obesity, hypertension, cardiovascular disease, respiratory disease, diabetes.

HIV/AIDS = Testing, treatment, monitoring; Women's and reproductive health = Family planning, antenatal care, cervical and breast cancer screening, screening and treatment of sexually transmitted infections; Tuberculosis = Testing, treatment, monitoring.

[a] The City of Johannesburg employs hundreds of individuals at the city and regional headquarters. We interviewed senior and mid-level officials.

[b] The denominator represents clinical staff who are present day-to-day, i.e. nursing staff.

[c] For service providers, this reflects actual service provision. For facility managers this reflects responsibility for service provision.

need to return to the clinic on different days for different services, stand in more than one queue when they visit, or see more than one healthcare provider during a visit. However, their descriptions of integration still implied that multiple providers might be required, depending on the timing of diagnosis or identification of multiple needs. This is illustrated by comments from one city/regional manager.

> "It does make more sense to treat a patient and HIV at the same time . . . so that patient does not have to queue. HIV integrates with chronic [care patients] and family planning, and TB is integrated with HIV. If you have HIV and then find out that you have TB, [the nurse] will send you to ART or TB. If chronic or family planning, you can be seen in TB room."
>
> –City/regional manager.

Most small primary health clinics in South Africa rely on nurses, usually with different levels of training, to provide care and treatment for all patients. Some older clinics have just one waiting area for patients; while newer clinics have designated areas for chronic diseases, acute care and maternal and child health. The respondents were unclear whether they felt integration meant that all staff should be able to provide all services, meaning that patients could wait anywhere and be seen by any clinical staff member, or whether patients would need to be directed to special queues or staff members for integrated care. In fact, most of the respondents did not address this structural aspect of integration at all when asked about what integration should look like in clinics. However, one city/regional manager noted that integration should require just one queue for any provider. Two city/regional managers said that integration could involve many queues for different primary health needs or concerns, but that nurses should then be able to manage all additional healthcare needs presented by their patients during the consultation.

**Table 2. Staffing and service at three public clinics in an administrative region of the city of Johannesburg, South Africa.**

| | Clinic 1 | Clinic 2 | Clinic 3 |
|---|---|---|---|
| Average number of patients seen per month | 3,906 | 5,050 | 4,678 |
| Staff employed at facility—total | 26 | 24 | 21 |
| Facility/operations manager | 1 | 1 | 1 |
| Doctor/registrar (visits occasionally/not located there) [a] | 2 | 2 | 2 |
| Dietician | 0 | 0 | 0 |
| Primary healthcare nurse[b] | 3 | 5 | 6 |
| Professional nurse[b] | 6 | 7 | 3 |
| Enrolled nurse/Nurse assistant[b] | 1 | 1 | 1 |
| Health advisor | 1 | 1 | 0 |
| Social worker | 1 | 0 | 0 |
| Counsellor/lay counsellor | 5 | 4 | 4 |
| Pharmacy/pharmacy assistant | 1 | 0 | 1 |
| Other administrative or operational staff[c] | 5 | 3 | 3 |
| Number of days per week that service is offered[d] | | | |
| HIV/AIDS | | | |
| Counselling and testing | 5 | 5 | 5 |
| ART initiation | 5 | 5 | 1 |
| Routine treatment (adults) | 5 | 5 | 4 |
| Routine treatment (children) | 5 | 5 | 1 |
| PMTCT | 5 | 5 | 4 |
| Tuberculosis | | | |
| Testing | 5 | 5 | 5 |
| Treatment (adults and children) | 5 | 5 | 5 |
| Women's/reproductive health | | | |
| Abortion | 0 | 0 | 0 |
| Breast & cervical cancer screening | 5 | 5 | 5 |
| Breast & cervical cancer treatment | 0 | 0 | 0 |
| Antenatal care | 5 | 5 | 5 |
| Family planning | 5 | 5 | 5 |
| Sexually transmitted infection (syndromic management) | 5 | 5 | 5 |
| Chronic disease management | 5 | 5 | 5 |
| Obesity | 0 | 5 | 5 |
| Diabetes | 5 | 5 | 5 |
| Hypertension | 5 | 5 | 5 |
| Cardiovascular disease | 5 | 0 | 5 |
| Respiratory disease | 5 | 5 | 5 |
| Immunizations/Child health | 5 | 5 | 5 |
| Mental health | 0 | 0 | 0 |
| Medical male circumcision | 0 | 0 | 0 |

[a] A "registrar" in South Africa is a residency position (i.e. operating under the supervision of a medical doctor).

[b] Generally training for the nurses is as follows: Enrolled nurse, 2 years; Professional nurse, 4 years; Primary Healthcare nurse, 6 years.

[c] Includes administrative assistants, data capturers, general workers and cleaners.

[d] Results are based on reports from the facility managers, and thus may represent normative schedules.

**When, how, and why integration of ART services occurred.** When asked why and when HIV treatment services were integrated into the services offered by public clinic staff, some respondents suggested that South Africa's National Department of Health had been planning to integrate a range of services, including HIV testing and treatment, within the clinics for some time. Some also mentioned that policy documents had stressed the need for integration "since 2004" or "since 2010." But it seems that HIV-related integration only really occurred when the standalone CCMTs (which were supported by local NGOs) were discontinued in 2013–2014.

When the CCMTs were operational, the public clinics and their staff were able to offer HIV testing but referred patients to the CCMTs for ART. The city/regional respondents noted that following the closure of CCMTs, nurses at public clinics in the region were required to offer the services they had been providing previously plus HIV treatment services, and that at first there were no adjustments to staffing or budgets. According to two respondents, integration as a policy was not planned for timeously, leaving some clinics unprepared. A regional manager described how the loss of CCMTs, and the staff assigned to them, necessitated reorganization of services.

> "And clearly, when the funding model changed and the core concept of . . . "technical support" kicked in, facilities had to now lose those resources, and clearly we were not, like, immediately capacitated. So you had to now see how best with your current staff you juggle around and deliver services."
>
> –City/regional manager.

Ultimately, the city/regional respondents noted that the Gauteng Provincial Department of Health (where the City of Johannesburg is located) did make additional funds available for clinics—following requests from the region for additional resources. However, the additional funds were reportedly still insufficient to make up for the initial loss of human and financial resources. This kind of *post hoc* planning was portrayed as commonplace in the public health sector.

> "Sometimes policy changes are announced politically before we are prepared. Certain things get announced over the media, and this puts pressure on us to make changes without the finer detail or resources. These are some of the challenges."
>
> –City/regional manager.

**Integration in practice—HIV.** Speaking generally regarding the model for and extent of HIV integration at the ten clinics in the region at the time of the interviews, four of the nine city/regional managers suggested that it "depends on the site". They indicated that at some clinics, nurses treated all chronic conditions, including HIV, during the same visit. In some cases, they even offered patients family planning and immunization for children during the same consultation. At other clinics, nurses reportedly had not yet been trained to offer all chronic care services and so were unable to provide an integrated service, but the three city/regional managers indicated that although there might be separate rooms for different conditions, patients would "be given one appointment for everything". One regional manager contradicted other reports, stating that HIV integration did not appear to be occurring; however it was unclear what kind of integration this person was referring to.

Reports from the clinic managers and healthcare providers supported the "it depends" perspective put forward by the city/regional managers regarding HIV service integration in the three study clinics. They further clarified that the "direction" of the integration might vary. At all clinics, counselling on HIV testing was reportedly offered in all consulting rooms. However, actual testing was often not offered during the same consultation. At clinic 1, staff reported that not all patients could obtain testing during their consultation for other services because "there is not enough time." Those patients wanting to test would be directed elsewhere in the clinic. At clinics 2 and 3, all patients wanting to test were referred to a specific area. One city/regional manager justified this approach saying that switching from condition to condition was "difficult" for staff.

> "What is happening, at first they wanted to—one nurse to see all different clients, but it's difficult. As a nurse in the consulting room you cannot say—see the client coming here and say[ing] I've got . . . hypertension. . .. Then . . . next, I'm pregnant. . . . Next, abdominal pain, headache, you know switching from [condition to condition]–it's difficult. So what we do, we have integrated, but . . . we divided them according to the conditions . . .[for example] we say all the babies that are coming for immunization will go to room 1."
>
> –City/regional manager.

HIV treatment was also not integrated into general primary care services, except TB services, meaning that patients who required HIV treatment had to wait for a nurse specializing in HIV treatment. Despite this seeming disadvantage for HIV-positive patients, it seemed that when seen by the HIV treatment nurse, there was an opportunity for integrated care. The HIV treatment nurse could reportedly offer chronic disease management and reproductive healthcare during the HIV treatment consultation (so primary care services were integrated into HIV treatment services).

Finally, some HIV-related services seem to have been integrated because of clearly overlapping protocols or specializations. For example, at clinic 1, if a female patient tested positive for HIV, the nurse administering the test would offer the patient a Pap smear and family planning at the same time. At clinic 2, the antenatal care nurse, who administers immunizations to children, also manages PMTCT patients. At clinic 3, the HIV nurse tests HIV patients for hypertension and provides Pap smears as part of her routine management of HIV-positive patients.

**Integration in practice—TB.** When asked about TB integration, four of the nine city/regional managers indicated that TB services were still "somewhat separate" from other services in the clinics in the region. According to the clinic managers and healthcare providers, all three of the study clinics had a TB-specific consulting room or area. TB screening was not spontaneously offered as part of other clinical services at any of the clinics; rather it was offered only to individuals exhibiting symptoms of TB. The level of integration of primary care services into TB services was also limited. At clinic 2, a TB nurse who managed and treated TB patients also offered chronic care and SRH services. However, this was not the case at clinics 1 and 3.

The respondents made distinctions between integration of HIV or TB services with primary care services and the integration of HIV and TB services with each other. One of the city/regional managers noted that at some clinics,

> "You find one nurse doing both services. And when you get to their files you'll find their TB cards, their ART cards, their blood results; everything in one pocket."
>
> –City/regional manager.

However, of the three study clinics, according to clinic managers and healthcare providers, just one had clearly integrated HIV and TB services saying that diagnosis of one condition prompted screening for the other. The other two clinics noted that the nurses responsible for TB service provision were lower level "enrolled nurses," and thus management of other conditions was not within their scope of practice.

## Barriers to integration

According to the study participants, there were a number of reasons why integration of HIV and TB services (with each other and into the clinic's broader offering of services) had not been fully implemented at the three study clinics. The most commonly reported barriers to integration were staff-related. Thirteen of the 19 participants noted that there was insufficient training for staff and that most staff were not "multi-skilled" and able to provide comprehensive services, including HIV and TB testing and treatment. It was noted that if nurses were relatively unskilled (e.g. "enrolled nurses" or newly graduated nurses), they were usually confined to offering just one or two service types. To illustrate, a healthcare provider at clinic 3 noted that TB services were not integrated with other services, "because you find out TB is done by an enrolled nurse. . .who can't do anything related to HIV".

When asked specifically about training, the clinic managers reported that some, not all of their nursing staff had training on HIV treatment and TB service provision. In fact, TB and HIV treatment were viewed as services requiring *extra* training, i.e. beyond their official education. However, this meant that staff who were able to offer HIV treatment and TB services had been trained on basic primary healthcare, including contraception provision.

Challenges regarding training and experience were said to be exacerbated by staff shortages, which were reported by nearly half of the respondents. There was a sense that more staff would help mitigate this problem. However, one city/regional manager added that clinics need more staff regardless of whether or not integration is introduced: "On a year to year basis, whether integration or not, you'll still need nurses".

A lack of sufficient space and other infrastructural challenges were also listed as barriers to integration by nearly two-thirds of the participants. According to the respondents, small consultation rooms with limited storage space could not accommodate all the medicines and consumables needed to manage all health conditions, which resulted in nurses having to fetch medicines from elsewhere in the clinic, presumably disrupting consultations and lengthening consultation times. There was also a limited number of consultation rooms, implying that even if the number of clinical staff were to increase, the clinic structure would not allow for more concurrent consultations to occur. One city/regional manager described the problem well:

> "The infrastructure fails us. Depending on the size of the clinic, you find that you want to provide services in a particular way, but the space area that you've got does not give you, you're not at liberty that you can do what you want to do. It limits you."
>
> –City/regional manager.

## Impact of integration

All but two of the 19 study respondents thought that the integration of HIV and other primary healthcare services was a good idea. They felt that integration would result in less "hassle." In fact, half of the respondents—mostly at the city/regional level—believed that improved

convenience for patients was a major benefit of integration. They reported that integration prevented patients from having to move around the clinic and from going to the clinic at different times or on different days to meet all of their healthcare needs.

According to four city/regional managers and three healthcare providers, integration promotes the "comprehensive" management of patients, which has many benefits. These might include improved management of co-morbidities, including TB, mitigation of side effects, and increased early detection and treatment of health conditions.

> "You know, the main thing with integration is that it gives the nurse an opportunity of knowing fully what you are dealing with."
>
> - City/regional manager.

Seven participants—mostly healthcare providers—spontaneously noted that they thought that integrating HIV and TB was a particularly good idea. They indicated that the two health conditions are usually found together and so integration encourages early detection and treatment of both.

> "For me TB and HIV, they are like twins, so they should all be treated in an integrated way. That nurse must be trained to manage a TB [client], to manage an ARV client and you maximize your resources. That's what we need to do."
>
> –City/regional manager.

Integrated, comprehensive approaches to care were also mentioned as a way to improve the patient-nurse relationship; though this was mentioned by just two respondents—one healthcare provider and one city/regional manager.

> "I think it does help the clients because if you come to a room, and you find a sister who is nice and open to you, you tend to relax and you speak out. . .. It makes you to be broad. . . . talking to the woman regarding other things and finding out about the status, and then it makes you, it brings them close to you, and they get to open up."
>
> - Healthcare provider.

Five city/regional managers also spoke about a reduction in stigma as one of the benefits of integration. Instead of patients being required to stand in separate queues if they are HIV-positive or infected with TB, as was previously the case, integration allows them to stand in the same queue as other patients. One respondent explained that mitigation of stigma associated with seeking care for HIV and TB encourages attendance at clinics:

> "You know with me integration it's good because you know, I'm looking at the part whereby other patients would be discriminated and would feel left out and would feel that stigma of being separated as those are TB, these are HIV, you see what I mean? . . . People won't come to the facilities because they will think that, hey, when I go there I'll be labelled, like I'm TB, I'm HIV, I'm whatever. So [with integration] . . . nobody will say you are TB."
>
> –City/regional manager.

Despite strong "pro-integration" sentiment among the respondents, there were some perceived limitations to its effectiveness and possibly negative aspects. Eleven respondents

cautioned that integration would only work in reality if the barriers to integration—namely staff shortages, inadequate skills levels, and space shortages—were mitigated.

There were also concerns about the impact of integration on waiting times and workloads. Although five respondents felt that integration decreased waiting time because patients did not need to stand in separate queues, seven participants noted that integration increased the time that nurses spend per consultation and the waiting time in the "integrated care queue." An additional six participants underscored that integration increased nurse workloads. They explained that having to do a range of activities—health promotion, screening, testing, and treating conditions—at every consultation would make nurses' jobs more stressful.

". . .patients needing all services being seen by one nurse—it is good for the patient, but the nurse is expected to see 40 patients a day."

–Clinic manager.

"It will be time consuming for a single nurse to offer more than one service at the same time."

–Clinic manager.

When specifically asked about integration of TB into primary care services, there were contrasting opinions. All three clinic managers supported the integration of TB care and treatment into primary care services, but the healthcare providers and city/regional managers had differing opinions. Just three of the seven healthcare providers and five of the nine city/regional managers supported the integration of TB into primary care services. Respondents in favor of integration reported a reduction in TB-related stigma, lower default rates, and earlier detection and treatment. Though, eight respondents highlighted that the integration of TB into primary care services required adequate infection control measures. In fact, all reservations regarding TB integration were due to concerns about the risk of infecting other patients with TB. The healthcare providers from one clinic were particularly concerned about babies and young children, who they said had "low" immune systems and so were especially vulnerable to TB infection. Four city/regional managers supported the integration of care for certain TB patients only. They expressed that patients with multi- and extensively-drug-resistant TB or newly diagnosed and untreated TB patients still displaying symptoms should be managed separately from other primary care patients. Directly in contrast to their thoughts on integration, some respondents suggested that TB patients should be managed in a separate, contained part of the clinic where, "TB patients should be treated for everything".

## Discussion

The managers and healthcare providers interviewed for this study were familiar with the rhetoric on integration of services and cited public policy documents supporting integration. However, they acknowledged that actual integration of HIV treatment services, TB management and primary healthcare had only really occurred as a result of funding transitions in their region. When asked to define integration in their setting there were different explanations and descriptions of practice, unsupported by any clear policy language.

Nonetheless, these individuals, who were responsible for implementation of integrated services, were generally supportive of integrated care. They cited the benefits of providing comprehensive care both for provider-patient relationships and patient outcomes. They noted that integration could reduce stigma. However, working on the "frontlines," they had practical insight on barriers to integration of services and real concerns about the day-to-day impact of

offering integrated care in constrained settings. The respondents noted several ground-level challenges, such as limited staffing and nurses' experience and tenure of service. They highlighted that with limited staffing, offering integrated care could still result in long waiting times, and a shift of the burden and "hassle" to nurses. They also noted that outdated, inadequate clinic infrastructure would result in nurses going from consulting room to consulting room to find the necessary supplies for a range of conditions.

This study is limited in terms of the number of respondents and their single geographic location. This constrains the transferability or generalizability of the findings. However, as for all qualitative research, which tends to have small sample sizes, the focus is not on creating generalizable results, but rather establishing validity with regard to the content explored. Content validity refers to whether the results as expressed adequately and comprehensively reflect the perspective of the population of interest [24]. The open-ended discussions in this study allowed for exploration of a range of issues. We also reached "saturation" during the interviews, meaning that no new concepts or ideas were emerging in subsequent interviews, which is a sign that our sample size was sufficiently large [27].

Further, our results are in line with existing literature on integration. Integration of health services can be done for various reasons and takes many forms. Horizontal integration—referring to services offered by one provider or within one facility via structured referrals [28], has been most commonly targeted in research exploring the impact of integration on health outcomes, efficiencies, and service delivery costs [29]. There is evidence of integration of HIV, TB, and sexual and reproductive health services resulting in improved clinical and public health outcomes [8,30–33]; however, data on efficiencies and cost-effectiveness are limited [29]. To some extent, data on efficiency gains resulting from integration of services are lacking because of the shifting nature of service delivery in real world settings.

In fact, much of the data on the success of integration efforts comes from carefully controlled study environments. In real-world settings, such as this study setting, where health systems are constrained by many factors, it has been acknowledged that a lack of clear policies and guidelines defining and describing integration goals, inadequate healthcare worker training and retention, and a lack of integration of higher-level functions, such as financing for care, are all barriers to successfully integrating healthcare services [34]. These findings have been mirrored in research in South Africa. Research on integration of HIV and TB services faulted a lack of leadership and poor healthcare worker training and supervision for "inadequate" integration of services [35]. For integration of HIV and sexual and reproductive health services, again a lack of clear policy guidance, poorly trained healthcare workers, and a weak health system with insufficient referral and monitoring systems have all been cited as barriers to integration [36,37].

In conclusion, variable definitions for and modes of implementing integrated services are presented as challenges in many evaluations of integrated service delivery. Yet, clinic operations, especially in small, busy facilities, can be variable due to infrastructure and staff limitations, ever-increasing demands for care, and evolving patients' needs. This study bolsters the idea that integration is multi-faceted and contingent on local, if not site-specific, factors. The respondents noted routine exceptions to general practice, possibly even recommended practice, which depended on staffing and preferences for patient management. In the future in South Africa and in other settings contending with health service reorganization, staff consultations and support prior to and throughout phase-in of services changes could contribute to improved understanding of operational requirements, including staff needs, and improved patient outcomes. Healthcare workers are ultimately responsible for providing care, and meaningful capacitation and engagement with them should be seen as critical to the success of interventions aimed at shifting standard practice.

## Supporting information

**S1 File.**
(PDF)

## Acknowledgments

The authors would like to extend their thanks to all of the respondents for their contributions to this work.

## Author Contributions

**Conceptualization:** Naomi Lince-Deroche, Lawrence Long.

**Data curation:** Naomi Lince-Deroche, Rahma Leuner, Sharon Kgowedi, Aneesa Moolla, Caroline Govathson, Takiyah White Ndwanya.

**Formal analysis:** Naomi Lince-Deroche, Rahma Leuner, Sharon Kgowedi, Sinethemba Madlala, Pertunia Manganye, Barbara Xhosa.

**Investigation:** Naomi Lince-Deroche, Rahma Leuner, Sharon Kgowedi, Caroline Govathson, Takiyah White Ndwanya.

**Methodology:** Naomi Lince-Deroche.

**Project administration:** Naomi Lince-Deroche, Rahma Leuner, Sharon Kgowedi, Caroline Govathson, Takiyah White Ndwanya.

**Supervision:** Naomi Lince-Deroche, Aneesa Moolla, Caroline Govathson.

**Writing – original draft:** Naomi Lince-Deroche, Rahma Leuner, Sharon Kgowedi, Aneesa Moolla, Sinethemba Madlala, Pertunia Manganye, Barbara Xhosa, Caroline Govathson.

**Writing – review & editing:** Naomi Lince-Deroche, Rahma Leuner, Aneesa Moolla, Takiyah White Ndwanya, Lawrence Long.

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
