## [Decision Letter · Decision Letter 0]

17 Sep 2019

PONE-D-19-17035

Voices from the front lines: A qualitative study of integration of HIV, tuberculosis, and primary healthcare services in Johannesburg, South Africa

PLOS ONE

Dear Ms. Lince-Deroche,

Thank you for submitting your manuscript to PLOS ONE. After careful consideration, we feel that it has merit but does not fully meet PLOS ONE’s publication criteria as it currently stands. Therefore, we invite you to submit a revised version of the manuscript that addresses the points raised during the review process.

Kindly address the minor suggestion made by the reviewer.

We would appreciate receiving your revised manuscript by Nov 01 2019 11:59PM. To enhance the reproducibility of your results, we recommend that if applicable you deposit your laboratory protocols in protocols.io, where a protocol can be assigned its own identifier (DOI) such that it can be cited independently in the future. For instructions see: http://journals.plos.org/plosone/s/submission-guidelines#loc-laboratory-protocols

We look forward to receiving your revised manuscript.

Kind regards,

Peter M. Mugo, B.Pharm, MSc, PhD

Academic Editor

PLOS ONE

Journal Requirements:

' This study was made possible by the generous support of the American people through the United States Agency for International Development (USAID), award number AID-674-A-12-00029. The contents are the responsibility of the Health Economics and Epidemiology Research Office, a Division of the Wits Health Consortium (Pty) Ltd and do not necessarily reflect the views of USAID or the United States Government.

LL received the grant.'

We note that you received funding from a commercial source: Wits Health Consortium (Pty) Ltd

Additional Editor Comments (if provided):

Reviewers' comments:

Reviewer's Responses to Questions

**Comments to the Author**

1. Is the manuscript technically sound, and do the data support the conclusions?

Reviewer #1: Yes

2. Has the statistical analysis been performed appropriately and rigorously? 

Reviewer #1: N/A

3. Have the authors made all data underlying the findings in their manuscript fully available?

Reviewer #1: No

4. Is the manuscript presented in an intelligible fashion and written in standard English?

Reviewer #1: Yes

5. Review Comments to the Author

Reviewer #1: This is indeed an interesting paper that adds value to the knowledge and current issues relating to integration of health services for HIV and TB.

Introduction, methods, results and discussion - The current version address all comments made previously.

Recommendation - Table 1 assist readers to have a clear understanding of the study population but Table 2 does not really add weight to the paper. It is therefore recommended that authors remove Table 2 and add a paragraph with information that authors perceive useful and critical in the description of the study population.

Overall impression - the study is worth being published given the level of infection in South Africa.

6. PLOS authors have the option to publish the peer review history of their article (what does this mean?). If published, this will include your full peer review and any attached files.

Reviewer #1: No

---

## [Author Response · Author response to Decision Letter 0]

6 Mar 2020

PONE-D-19-17035

Voices from the front lines: A qualitative study of integration of HIV, tuberculosis, and primary healthcare services in Johannesburg, South Africa

Response to reviewers

Comments received from the editor and two reviewers are listed below. We have numbered them for ease in cross-referencing. Note that the lines numbers represent the track-changed version. 

Responses follow each comment in bold text. 

None.

Journal Requirements:

Done.

As noted in communication for this manuscript LAST YEAR: the data are considered restricted by the local ethics committee: the Health Research Ethics Committee (medical) at the University of the Witwatersrand. The study was protocol M140104. Permission to use the data can be obtained through a request to the Committee. Contact details are as follows:

Research Office Secretariat

Faculty of Health Sciences

Phillip Tobias Building, 3rd Floor, Office 301

29 Princess of Wales Terrace

Parktown 2193 

Johannesburg, South Africa

Telephone: +27 (0)11 717 1252

hrec-medical.researchoffice@wits.ac.za

www.wits.ac.za/research/about-our-research/ethics-and-research-integrity/

The information is now provided in the revised cover letter.

' This study was made possible by the generous support of the American people through the United States Agency for International Development (USAID), award number AID-674-A-12-00029. The contents are the responsibility of the Health Economics and Epidemiology Research Office, a Division of the Wits Health Consortium (Pty) Ltd and do not necessarily reflect the views of USAID or the United States Government.

LL received the grant.'

We note that you received funding from a commercial source: Wits Health Consortium (Pty) Ltd

As noted in communication for this manuscript LAST YEAR: The Wits Health Consortium, (Pty) Ltd is a division of the University of the Witwatersrand. It was established as a private, not-for-profit entity with the intention of providing the controls required to receive funding from groups such as USAID. The Health Economics and Epidemiology Research Office is a division within the Wits Health Consortium. Thus, it is the place of employment for the authors. It is not a funding source.

Additional Editor Comments (if provided):

Reviewers' comments:

Reviewer's Responses to Questions

Comments to the Author

1. Is the manuscript technically sound, and do the data support the conclusions?

Reviewer #1: Yes

Thank you.

2. Has the statistical analysis been performed appropriately and rigorously?

Reviewer #1: N/A

3. Have the authors made all data underlying the findings in their manuscript fully available?

Reviewer #1: No

See above.

4. Is the manuscript presented in an intelligible fashion and written in standard English?

Reviewer #1: Yes

Thank you.

5. Review Comments to the Author

Reviewer #1: This is indeed an interesting paper that adds value to the knowledge and current issues relating to integration of health services for HIV and TB.

Introduction, methods, results and discussion - The current version address all comments made previously.

Thank you. 

Recommendation - Table 1 assist readers to have a clear understanding of the study population but Table 2 does not really add weight to the paper. It is therefore recommended that authors remove Table 2 and add a paragraph with information that authors perceive useful and critical in the description of the study population.

We feel that Table 2 provides the reader with an at-a-glace view of the range of SRH, HIV and TB services provided at the facilities and that this is an important aid when reading through the presentation of the challenges, etc. with integration of care. Table 2 also provides a sense of the volume of patients and the number of staff managing them. We feel that this information could not all be converted to text, and that it is important as a reference.

Overall impression - the study is worth being published given the level of infection in South Africa.

Thank you.

6. PLOS authors have the option to publish the peer review history of their article (what does this mean?). If published, this will include your full peer review and any attached files.

Do you want your identity to be public for this peer review? For information about this choice, including consent withdrawal, please see our Privacy Policy.

Reviewer #1: No

There are no figures in this manuscript.

---

## [Editor Report · Decision Letter 1]

11 Mar 2020

Voices from the front lines: A qualitative study of integration of HIV, tuberculosis, and primary healthcare services in Johannesburg, South Africa

PONE-D-19-17035R1

Dear Dr. Lince-Deroche,

We are pleased to inform you that your manuscript has been judged scientifically suitable for publication and will be formally accepted for publication once it complies with all outstanding technical requirements.

With kind regards,

Peter M. Mugo, B.Pharm, MSc, PhD

Academic Editor

PLOS ONE
---

## [Editor Report · Acceptance letter]

28 Sep 2020

PONE-D-19-17035R1 

Voices from the front lines: A qualitative study of integration of HIV, tuberculosis, and primary healthcare services in Johannesburg, South Africa 

Dear Dr. Lince-Deroche:

I'm pleased to inform you that your manuscript has been deemed suitable for publication in PLOS ONE. Congratulations! Your manuscript is now with our production department. 

Kind regards, 

on behalf of

Dr. Peter M. Mugo 

Academic Editor

PLOS ONE